# Climate solution or corporate co-optation? US and Canadian publics' views on agricultural gene editing

**Sara Nawaz**¤*, **Terre Satterfield**

Institute for Resources, Environment and Sustainability, University of British Columbia, Vancouver, BC, Canada

¤ Current address: Institute for Science, Innovation and Society, University of Oxford, Oxford, United Kingdom

* sara.a.nawaz@gmail.com

## Abstract

The dexterity and affordability of gene-editing technologies promise wide-ranging applications in agriculture. Aiming to take advantage of this, proponents emphasize benefits such as the climate-mitigating promises of gene editing. Critics, on the other hand, argue that gene editing will perpetuate industrialized forms of agriculture and its concomitant environmental and social problems. Across a representative sample of US and Canadian residents (n = 1478), we investigate public views and perceptions of agricultural gene editing. We advance existing survey-based studies, which tend to focus on whether knowledge, familiarity, trust, or perceptions of naturalness predict views on gene editing. Instead, we examine whether broader societal concerns about industrialized food systems—a key claim about genetic engineering launched by critics—predicts comfort with gene editing. We also explore the predictive power of views of climate change as an urgent problem, following proponent arguments. Survey results explore gene editing views in reference to specific cases (e.g., drought-tolerant wheat) and specific alternatives (e.g., versus pesticide use). We find that people critical of industrialized food systems were most likely to express overall absolute opposition to the technology, whereas those concerned with the imminence of climate change were more likely to support climate-relevant gene editing. Our findings suggest the need for further research into the conditions upon which public groups find gene editing compelling or not—namely, if applications enhance or counter industrial food systems, or offer particular climate adaptive benefits. Furthermore, we argue that attention to broader societal priorities in surveys of perceptions may help address calls for responsible research and innovation as concerns gene editing.

## 1. Introduction

Following decades of debate, genetically modified organisms (GMOs) continue to be widely controversial. Now, with gene-editing technologies on the horizon—and in some cases,

**Data Availability Statement:** Anonymized data are available at: https://github.com/sara-nawaz/gene-editing-survey.

**Funding:** Genome British Columbia SOC005.

**Competing interests:** The authors have declared that no competing interests exist.

already in grocery stores—debates about GMOs (or, transgenic organisms produced using recombinant DNA technology) are redoubling. Proponents assert that gene editing is less expensive and less risky and controversial than genetic modification (GM), as it allows for precise edits, smaller deletions or additions, and the production of non-transgenic edited organisms [1–4]. One of the key benefits of gene editing, they argue, is its ability to be readily applied to a range of agricultural challenges—such as helping agriculture adapt to rapidly changing climatic conditions [5]. On the other hand, critics have raised myriad social, political, and cultural concerns about GMOs, and increasingly, apply these to gene editing [6–9]. Critics question, in particular, assumptions about the kind of food systems most desirable for the future—pitting, as pejorative, industrialized systems designed to produce food for profit against food systems that reflect more diverse social and ecological goals.

Just how foundational these divisions are, however, is unclear. Are public groups convinced, as proponents of gene editing would assert, by claims about climatic benefits? And/or are they sympathetic to the broader political economic and social critiques of genetic engineering? In this paper, we first review and reflect on past work on public risk perceptions and suggest that political economic critiques of industrial agriculture—historically excluded from the risk perceptions literature—may prove helpful in understanding current views on GE. Second, we consider whether these critical perspectives on industrial agriculture predict strong or unwavering opposition to gene editing. Finally, we consider a key assumption made by proponents about public views on gene editing, namely that public groups are more likely to support it if they are concerned about climate change. Detailed discussion of these lines of exploration follows, along with testable propositions for each.

## 1.1. Testing critics' arguments: Bringing political-economic systems into the study of risk perceptions

Research on public risk perceptions has long sought to understand what might motivate, or at least predict, views on biotechnology. Attitudes towards GMOs have been predicted by (mis) trust [10–14], perceived risks to individual and societal health, and economic, political and environmental concerns [15–19]. Knowledge and familiarity [14, 20, 21] and demographic characteristics such as gender, age, education, ethnicity and income [13, 22] have been found to be predictive as are the specifics of the application itself (e.g., whether applications involve plants vs. animals [12]). Lastly, political worldviews including judgments about equality and social order (e.g., egalitarian v. hierarchical views) have also been key in explaining risk judgments [23–25].

A parallel body of work also indicates important variables missing from the risk perceptions literature—variables that might also help explain views on biotechnology. Scholarship has highlighted political economic critiques of GMOs—paying discrete attention to a technology's control by corporations [26, 27] as well as linked ownership and/or intellectual property concerns [28–30]. Scholars have also questioned how GMOs have served the continuance of industrial agricultural and its reliance on large monocultures and external inputs such as pesticides [31, 32]. This body of work also asks whether GMOs are truly necessary and questions whether preferential funding has made such approaches successful in the first place [27, 33]. Scholarship has highlighted that more agroecological applications of GM were effectively locked out, ensuring a technology almost exclusively deployed in industrialized agriculture, with extensive reliance on monocultural production and petrochemical-based inputs [34, 35]. Together, these studies articulate a critique of GMOs rooted in an indictment of the political economic structures on which these technologies rely on and, arguably, perpetuate.

Initial studies of gene editing have indicated that views about this new class of technologies likely echo those expressed in the perceived risk of GMOs literature (with the former perhaps viewed slightly more favorably [36–38]). However, emerging studies continue to confine most explanatory variables to trust, knowledge, demographics and the perception of social ecological or economic risks and benefits of the technology itself. An exception is the recent study commissioned by the UK's Royal Society, which offers tentative evidence that system-level political economic concerns also arise with regard to gene-edited organisms. Drawing upon workshops with public groups, the study found several similar factors as relevant to people's views on the acceptability or unacceptability of gene-edited products. Participants viewed applications as unacceptable if, amongst other factors, they created monocultures and if applications prioritized individual and/or corporate wealth [39].

In sum, despite advances in deliberative or small group designs as noted above, the survey-based risk perceptions literature has under-represented beliefs about political economic systems. These beliefs extend beyond the individual-level or product-level risk items and instead involve questions such as: what kind of future food system is desirable [40]? How might a given technology or application support this or not? Paying attention to such systems-level questions highlights the differences between specific technological approaches, and the broader context in which they arise. Furthermore, incorporation of these factors into the study of risk perceptions—a critical field for influencing regulation and policy—might help shift the conversation on gene editing toward better incorporation of public values and priorities, as scholars of responsible innovation have long hoped [40–42].

We propose, then, that there is a need to better integrate broader questions about societal and systemic challenges facing food systems, and to explore how or whether people's views on such topics explain their views on gene editing. More specifically, we propose that people's attitudes and priorities with regard to the role of industrial agriculture in future food systems are relevant to their views on gene editing. While a survey cannot offer conclusions on what *causes* views of gene editing, it can begin to shed light on relationships between desirable political economic systems, and views on gene editing. To our knowledge, few studies of perceived risk have sought to explore these views. Exceptions include work by Amin and colleagues, who explored attitudes towards corporations vis-à-vis positions on biotechnology (in general, and GM salmon in particular) [43]. Beyond this, we are unaware of other efforts to apply such critiques of current, industrial food systems to survey-based perceptions work on gene editing or other genetic engineering technologies.

Because views about political economic systems are always broadly stated, we have opted to explore the salience of such critiques in quantitative perceptions studies by operationalizing views on the Green Revolution. Why the Green Revolution? Today, genomics-based approaches, including gene editing, are being heralded as a new Green Revolution, and many contemporary debates about genomics-based approaches and biotechnology appear fundamentally similar to those characteristic of debates about the Green Revolution [44, 45]. We hypothesize that the Green Revolution might thus serve as a viable proxy for understanding how people think about large-scale biotechnological transformations emerging in agriculture. The Green Revolution was a period of agricultural transition from 1940s-70s originating in Mexico and then India involving the use of high-yielding varieties, fertilizers and pesticides, irrigation, and mechanization [45, 46]. Few technologies have been as influential, or as eventually controversial, as were these. Arguments for the Green Revolution include its role in significantly reducing poverty, lowering food prices, rapidly increasing agricultural production, saving many lives at a scale that could not have been achieved by any other means, and converting India from an aid-dependent country to a net producing one [46]. Vehement critics and social movements in opposition to the Green Revolution are also widespread. For critics, the Green Revolution conceptualized problems of hunger in a manner that eventually

bolstered the capitalist, liberal state [45]. Equally pervasive are views of the Green Revolution as a set of interventions that deepened poverty, enabled the persistence of food insecurity to persist, and generated myriad environmental impacts [46].

We thus postulate that,

*P1: Participants with more negative views about corporations and the Green Revolution—that is, those that express critiques of current industrialized food systems—will have more negative views of gene editing.*

## 1.2. Testing critics arguments, part 2: Exploring resistance to tradeoffs

Our second proposition seeks to explain strong, even absolute, opposition to gene editing. While research has noted the social, political, or moral reasons that people might be skeptical of genetic engineering [e.g., 47], proponents of genetic engineering (such as agricultural companies) have often assumed that public groups will be persuaded to accept these technologies if they are made aware of their full array of benefits, taking as a given that opposition is based on a lack of knowledge, or on strict moral views [12, 48]. In studying morally laden topics more generally, some researchers have sought to understand when and why people refuse to make a choice of any kind between two options; a key insight from this body of work is that people 'opt out' or refuse to make a choice or consider a trade-off when they deem the question morally repugnant. These are generally known as 'taboo trade-offs' wherein the idea of choosing is itself rejected outright [49–53].

We propose that some publics may, in such a way, view decisions about gene editing as a taboo or as an intolerable proposition. Indeed, a 2016 study on US publics' perceptions of GMOs [53] found that 45% of their sample were absolutely opposed to GM under any conditions. For these individuals, GM violated 'basic moral principles' and thus was unacceptable, regardless of its consequences, good or bad. The authors found that such absolutist moral opposition was predicted by a sense of disgust. While the authors' interpretation has been contested [54], such perspectives align with broader claims by proponents of GMOs, who have a tendency to regard opposition to genetic engineering as irrational resistance that is divorced from tangible risks and benefits [55].

Given the broader critiques of genetic engineering discussed above, we wonder whether such strong or absolutist opposition might be rooted not in irrationality, but rather in political beliefs about the role of industrial agriculture in future food systems. To test this, we first ask: to what extent do people express outright opposition to gene editing? Do some people simply refuse to answer, and so 'opt out' of the question entirely when asked to state their preference for gene editing in the context of trade-offs involving specific benefits (e.g., accepting gene editing in order to reduce pesticide use)? Furthermore, do critical views on industrial food systems predict such outright opposition if and when it occurs?

We thus propose that:

*P2: Critical attitudes about industrialized food systems (as measured through views on corporations and the Green Revolution) will predict the likelihood of rejecting any need to declare a choice or preference in the context of a relevant trade-off.*

## 1.3. Testing proponents' arguments: Assessing the effect of perceived urgency of climate change

In addition to exploring the potential predictive power of critical attitudes toward industrialized food systems vis-à-vis the perceived risk of gene editing or the likelihood of more absolute positions (e.g., tradeoff rejection), we also want to understand how people evaluate one of the

key claims proponents are making about gene editing: that it is crucial for addressing climate change [56, 57]. According to this line of argument, agriculture both drives climate change (contributing between 21% and 37% of greenhouse gas emissions [58]), and conversely, climate change will increasingly affect agriculture via a variety of stressors such as more frequent droughts, intensity and timing of sunshine and rainfall, lowland flooding, waterlogging, evapotranspiration, and increased pests and diseases [59]. While most GM products currently on the market are staple crops developed to be either herbicide- or pest-resistant, proponents argue that new editing technologies offer a much wider variety of potential applications. Proponents—including technology developers, agricultural companies, and policymakers, amongst others—argue that gene editing will help farmers keep pace with rapidly changing climatic conditions by providing more resilient species [60] and crops that are drought- [61], heat- [62], and salt-tolerant [63]. Researchers may even develop non-leguminous crops with nitrogen-fixing abilities [64]. Gene editing could also reduce waste in the food cycle (e.g., non-browning apples [65]), a problem that contributes indirectly to climate change, by increasing the total land needed for or converted to agricultural production.

While scientists studying these applications have proposed such climate-adaptive uses for gene editing, it is not clear whether such possibilities are persuasive or with whom. While views on climate change have been demonstrated to affect attitudes toward novel and emerging technological approaches, such as assisted migration or GMOs in forestry [66, 67] and carbon dioxide removal technologies [68], minimal research has demonstrated the relationship between views on climate change and views on biotechnology generally, let alone on gene editing.

Our final proposition thus explores whether views about the urgency of climate change predict attitudes toward gene editing. To test this, we propose the following proposition:

> P3: Perceiving climate change as an urgent problem will predict openness to applications of gene editing that involve climate-adaptive agriculture.

Altogether, these three propositions allow us to understand (1) the importance of views about food systems for predicting views of gene editing applications, particularly strong opposition to gene editing, (2) whether another system-level concern—namely, the urgency of climate effects—drives opposition to gene editing, and (3) whether proponent claims about climatic benefits of gene editing resonate with public groups.

## 2. Methods

We employed an online survey to explore these questions in detail. Data were collected using Qualtrics software (https://www.qualtrics.com). We distributed the online survey to adults over the age of 18 living in Canada and the United States. The study was approved by the University of British Columbia's Behavioural Research Ethics Board (ID number H18-03066). Participants were informed that continuing with the survey would indicate their consent to participate (i.e., written consent). We used a digital data collection company (Dynata, https://www.dynata.com) to generate a sample stratified by age and gender across each state or province. After removing incomplete responses and those that were completed in less than 5 minutes, the final analyzed sample was n = 1478. The median completion time for analysed surveys was approximately 13 minutes.

### 2.1. Dependent variables

We used two sets of dependent variables (DVs) to measure attitudes towards gene editing in agriculture. The first (DV1) elicited judgements of comfort through discomfort across three

cases of gene editing. The cases were selected to include a range of organisms, techniques involved, and purposes. These included: (1) tomatoes edited to return heirloom sweetness traits that had been lost in prior breeding processes, (2) cattle edited to be more likely to pass on traits for 'hornless-ness' to offspring, thereby removing the need for painful dehorning surgeries, and (3) wheat edited to be climate-resilient (e.g., drought-resistant). Participants were asked to rate their comfort with each application. We used comfort instead of acceptance, as this was both a more global and intuitive measure, as well as more suitable to thinking about the 'idea' of something, given that these are often technologies that have been proposed but are not yet widely used.

We also sought to explore attitudes towards gene editing via a second variable (DV2), which addressed trade-offs involving specific benefits and risks. We sought to understand how the context of particular trade-offs affected participants' responses. Each survey participant received two vignettes describing two types of potential benefits of gene editing. The benefits, respectively, were a reduction in the use of pesticides (an environmental harm often identified by food system critics) and increases in biodiversity (given less land converted to agriculture). A Likert-style scale measured preferring through not preferring gene editing in exchange for these specific benefits. In other words, participants were offered variations of choices such as (1) a reduction in pesticide use in exchange for greater use of editing technologies, and (2) lessened biodiversity loss due in exchange for use of gene editing. When eliciting these responses, we provided participants the option to 'opt out' of the trade-off between benefits and use of gene editing, as we anticipated many might reject the need for such trade-offs in the first place.

Given the importance of understanding the different technologies in question vis-à-vis each other and GM, we included at the outset of the survey a question assessing participants' familiarity with various genetic engineering concepts (e.g., GM and gene editing), followed by a set of brief explanations (i.e., one sentence each) defining these. We also asked participants whether these three items appeared similar or different to each other, both in the form of a multiple choice and free response option to note any reflections or questions. Responses indicated, unsurprisingly, that respondents were more familiar with GM than other methods. Free response answers did, however, appear to suggest a high degree of comprehension of the definitions provided.

## 2.2. Explanatory variables

We included a set of explanatory variables in the survey in order to explore their predictive power vis-à-vis the above dependent variables. Key established variables were also included so as to ensure model robustness: these included demographic and attitudinal variables, namely gender, age, race/ethnicity, education, political orientation, religiosity, and income. Also included was an established attitudinal trust scale [7] that we adapted to evaluate trust in regulators, government officials, and scientists.

We also included new and/or altered attitudinal scales in order to explore the propositions listed above. We adapted existing climate attitude scales, drawing inspiration from prior work [69–73] to introduce a scale that explored a sense of urgency (or ambivalence) about climate change. As discussed above, we introduced a scale on corporate criticism (degree of criticism of corporate monopoly and power) based on an existing scale [46]. Lastly, we developed a new scale to capture attitudes toward the Green Revolution, based on academic investigations and claims as to the societal benefits and costs of the Green Revolution [45, 46]. For this scale, participants were provided a short tutorial on the Green Revolution before being asked to respond to these claims, which included optimistic statements such as:

• "The Green Revolution was a positive development for farmers in countries like India".

- "Because of the Green Revolution, many fewer people starved or suffered hunger than otherwise would have".

- "The Green Revolution brought much-needed increases in agricultural productivity".

  More critical views about the Green Revolution were captured by items such as:

- "The Green Revolution led to big losses of traditional crops & agricultural biodiversity".

- "The Green Revolution has exacerbated inequalities amongst farmers".

- "The Green Revolution was not necessary; such advances in productivity could have occurred in a more environmentally sustainable manner".

- "The Green Revolution has contributed to the excessive use of pesticides and fertilizers in modern farming".

We included a knowledge question in order to control for lack of familiarity with the Green Revolution: "I was familiar with the meaning of the term "the Green Revolution" before reading this question".

Response options for all scales were: "Strongly disagree", "disagree", "neither agree nor disagree", "agree", "strongly agree", and "don't know/not sure".

## 2.3. Statistical analyses

Statistical analyses were performed in R (version 4.0.3). First, we conducted an exploratory factor analysis on the items in each of the attitudinal scales, determining which scale items to retain for analysis. We utilized several criteria to judge the suitability of attitudinal scales for factor analysis: first, we conducted Shapiro-Wilk tests to check that data are normally distributed [74]; second, we conducted Kaiser-Myers-Olkin tests to check the sampling adequacy [75]; third, we checked the correlation matrices and conducted Bartlett sphericity tests to check that factor analysis was appropriate [76]. All scales were deemed suitable according to the criteria for each of these tests. We determined the number of factors for each scale using a combination of the following: Kaiser's rule (eigenvalues must be greater than 1) and parallel analysis [77]. We utilized varimax rotation in order to facilitate ease of interpretation of factor analysis results. We removed items that did not load above 0.4 on any factor, or that lowered the overall Cronbach's alpha score of the overall factor. We then checked the correlations of all variables, to confirm relative independence of our predictors.

We next conducted ordered logistic regressions on both dependent variable questions. Ordered logistic regressions are standard for analyzing survey questions that have discrete but ordered responses [78]. The first set of regression analyses used ordered logistic regressions to explore DV1, which introduced survey participants to three cases of gene editing and asked them to rate their discomfort through comfort with each. The second set of regression analyses also used ordered logistic regression to explore DV2, which asked participants to rate their rejection or acceptance of gene editing in the context of trade-offs involving specific benefits and risks. The third set of regression analyses used binomial logistic regressions to explore DV2, comparing those who either moderately or strongly preferred gene editing, with those who opted out of the trade-off altogether. For all regressions, we checked for multicollinearity again by confirming that VIF scores for each variable were well under 10.

## 3. Results

The results here describe our evaluation of the propositions described above, regarding (1) the predictive power of critiques of industrial agriculture on attitudes towards gene editing; (2) the

predictive power of such views with regards to opting out of, or absolutely rejecting, any trade-offs involving gene editing; and (3) the predictive power of climate change attitudes as they concern openness to gene editing.

## 3.1. Explanatory variables

To examine all three propositions (P1-P3), we began by first factor analyzing results for the attitudinal scales. Table 1 describes these analyses, and highlights five scales evaluated as predictors in the regression analyses to follow: (1) trust in relevant institutions (a strong predictor in earlier studies referenced above [10–14]), (2) criticism of corporations, (3) attitudes towards climate change, (4) critical views about the Green Revolution, and (5) optimistic views regarding the Green Revolution. While we do not have a proposition about trust, we highlight it in the results discussion, as it is long and well-established in the literature, has been a particularly significant predictor, and offers a useful comparison for the other more novel variables we tested.

A few notes about the Green Revolution scales: as noted above and prior to the Green Revolution questions, we also included a question to assess familiarity with the Green Revolution. We conducted our analyses two ways: (1) excluding those participants who were unfamiliar with the Green Revolution, and (2) including everyone. Results were similar, which may indicate that the survey's simple description of the Green Revolution as techniques "including pesticides and fertilizers" was enough information to gauge participants' general attitudes on the topic of modern farming approaches. The results presented here include all participants' responses, regardless of their prior knowledge about the Green Revolution. Second, we have included two separate Green Revolution scales for several reasons. The 'criticism' scale reflects some of the key critiques or objections to the Green Revolution, and the 'optimism' scale

**Table 1. Factor-analyzed results of attitudinal scales.**

| Attitudinal scale | Items and loadings | α | Variance explained |
|---|---|---|---|
| **Trust** | I trust regulators to make sure the risks of genetic technologies are minimized (0.83) | 0.86 | 0.46 |
| | I trust scientists to adequately manage the risks associated with genetic technologies (0.82) | | |
| | I trust agricultural companies to be conscious of their responsibilities in using genetic technologies (0.81) | | |
| **Corporate criticism** | The increasing influence of large corporations is a problem (0.72) | 0.71 | 0.34 |
| | Globalization has positive impacts for the large majority of people (0.65) | | |
| | I understand that corporations try to make money, but I don't think they should control knowledge through patents (0.63) | | |
| **Climate ambivalence** | Scientists agree that the evidence for human-caused climate change is partial at best (0.70) | 0.69 | 0.19 |
| | The unique problems of climate change necessitate more caution than action (0.64) | | |
| | Many other problems that also impact people globally are more urgent than climate change (0.60) | | |
| **Green Revolution criticism** | The Green Revolution led to big losses of traditional crops & agricultural biodiversity (0.80) | 0.76 | 0.26 |
| | The Green Revolution has exacerbated inequalities amongst farmers (0.65) | | |
| | The Green Revolution has contributed to the excessive use of pesticides and fertilizers in modern farming (0.62) | | |
| | The Green Revolution was not necessary; such advances in productivity could have occurred in a more environmentally sustainable manner (0.60) | | |
| **Green Revolution optimism** | The Green Revolution brought much-needed increases in agricultural productivity (0.81) | 0.8 | 0.26 |
| | The Green Revolution was a positive development for farmers in countries like India (0.74) | | |
| | Because of the Green Revolution, many fewer people starved or suffered hunger than otherwise would have (0.71) | | |

Item responses were: "strongly disagree", "disagree", "neutral", "agree", "strongly agree". "Don't know/not sure" response options were provided but have been excluded from analysis.

reflects some of the key justifications or apologies for the Green Revolution. There are some topical differences: the 'criticism' scale pertains more to questions of impacts to biodiversity and traditional crops, exacerbation of inequality, environmental sustainability, and use of pesticides and fertilizers, while the 'optimism' scale pertains more to yields, hunger, and productivity. We retained both scales as a way to account for the possibility that some might agree with the criticisms, but *also* agree with statements of optimism about the Green Revolution.

Before proceeding to the regression analysis, we checked correlations amongst all explanatory variables. Political conservativism correlated negatively with climate ambivalence (correlation of 0.28), unsurprisingly. Attitudes of Green Revolution criticism and corporate criticism correlated with each other (0.39), indicating a degree of overlap across these critiques of industrialized agriculture. Views on Green Revolution optimism and trust correlated with each other as well (0.32).

### 3.2. Predicting comfort/discomfort with gene editing (P1 & P3)

To evaluate P1, we introduced an application that involved the use of gene editing to create more drought-resistant wheat. We found that participants who were less ambivalent and more certain about the urgency of climate change as a problem were more likely to be comfortable with the application. This finding is shown below in Fig 1. We conducted the same analysis for two other applications that were less climate-relevant: (1) tomatoes edited to have heirloom sweetness traits lost during the breeding process, and (2) cattle bred to be hornless so as to avoid painful dehorning processes. Results were similar across all three cases, with a few exceptions. In the tomato case, being familiar with gene editing predicted that a participant would be more comfortable with the application. In the cattle case, being male predicted comfort with the application. Crucially, climate ambivalence did *not* predict discomfort in either of those two cases that were not presented as having clear climate-related benefits.

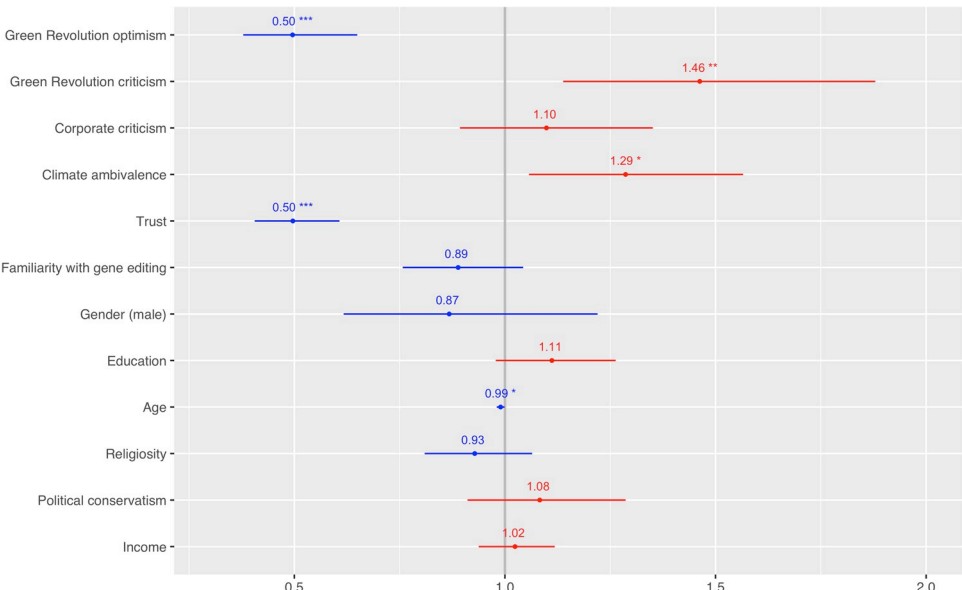

**Fig 1. Discomfort with drought-resistant wheat.** Plotted below are results of ordered logistic regressions on participants' discomfort with a specific application of gene-edited wheat. Odds ratios represent the odds than an outcome will occur given a specific variable. The significance codes for P values are: 0 '***' 0.001 '**' 0.01 '*' 0.05. Participants were less likely to be uncomfortable (more likely to be comfortable) with drought-resistant wheat if they were optimistic about the Green Revolution, were less critical of the Green Revolution, did not express ambivalence about climate change, did not express criticism of the Green Revolution, expressed higher levels of trust, or were older.

Alongside the commonly cited predictor of trust (and familiarity, in the tomato case), we found evidence that critiques of industrialized agriculture predicted preferences as postulated in P1, with participants' sense of optimism about the Green Revolution predicting comfort with all three applications. Criticism of the Green Revolution, however, only predicted attitudes with the wheat application, not with either the tomato or cattle applications. Attitudes regarding corporations were not statistically significant predictors of attitudes towards any of the applications studied.

## 3.3. Predicting preferences for gene editing, in the context of specific tradeoffs (P1 & P3)

Fig 2. below shows predictors of gene editing in the context of trade-offs involving pesticide use and biodiversity. With regards to P3, results indicate that climate change attitudes were significant predictors of preferences for gene-editing when traded off against both types of purported benefits (reduced pesticide use or reduced loss of biodiversity. A sense of ambivalence about climate change predicted preferences for both increased pesticide use and greater biodiversity loss, as opposed to greater use of gene editing.

With regard to P2, we found that critiques of industrial agriculture explains preferences for gene editing in relation to specific benefits. Namely, in the context of trade-offs involving both gene editing as preferable to pesticide use, and gene editing as preferable to biodiversity loss, we found that criticism of corporations predicted opposition to gene editing. Green Revolution optimism, but not criticism, predicted preferences for gene editing in both tradeoff contexts (e.g., in exchange for biodiversity gains and pesticide reduction).

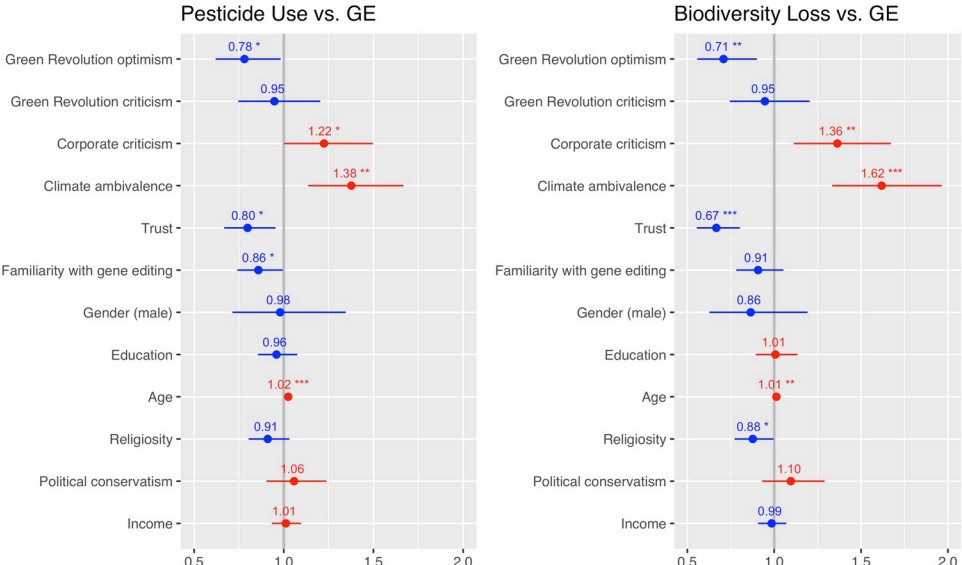

**Fig 2. Preferences for increased pesticide use or biodiversity loss, as opposed to gene editing.** Plotted below are results from ordered logistic regression on the likelihood of preferring (1) increased pesticide use (vs. gene editing), and (2) greater biodiversity loss (vs. gene editing). This analysis excluded those who 'opted out' of each of the trade-offs, presenting only the findings relating to those who answered the two trade-off questions. Odds ratios represent the odds than an outcome will occur given a specific variable. Confidence intervals (2.5% to 97.5%) offer a range of plausible odds ratios for each of the independent variables. The significance codes for P values are: 0 '***' 0.001 '**' 0.01 '*' 0.05. Participants were more likely to prefer either increased pesticide use or greater biodiversity loss over gene editing if they were critical of corporations, more ambivalent about climate change, and older. They were more likely to prefer gene editing over an increase in pesticide use or biodiversity loss if they were optimistic about the Green Revolution and expressed higher levels of trust. Participants were also more likely to prefer pesticide use if familiar with gene editing, and biodiversity loss if they were less religious.

### 3.4. Predicting the likelihood of *opting out* of the pesticide and biodiversity trade-offs (P2)

Regarding P2, we found that a sizeable number of participants opted out of the tradeoff question—approximately 29% for the pesticide tradeoff, and 24% for the biodiversity tradeoff. Drawing on forthcoming qualitative work [79], we offered participants two possible reasons for their decision to opt out, and found that, for each of the tradeoffs, 36% of those opting out selected a lack of information on "who owns and controls these technologies" as the motivating reason for opting out, and 64% selected a lack of consideration of alternatives: "other ways of avoiding pesticide use/conserving biodiversity".

Comparing those who were supportive of gene editing with those who opted out of the tradeoff entirely (shown in Fig 3) indicates that both criticism of and a lack of optimism about the Green Revolution predicted likelihood of opting out of both the trade-offs involving gene editing. Counter to our proposition, however, critical views on corporations did not predict who might be more likely to opt out of the trade-offs.

## 4. Discussion

Primary observations across our three propositions can be summarized as follows: we found evidence that (1) optimism about the Green Revolution (which we interpret as optimism about industrialized agriculture) predicted both comfort with gene editing (DV1) and preferences for gene editing in the context of key tradeoffs (DV2); (2) optimism about the Green Revolution indicated a lower likelihood of opting out of or avoiding a tradeoff, whereas pessimism around the Green Revolution indicated a higher likelihood of opting out of the request to participate in a tradeoff in the first place; and (3) those who were more certain about the

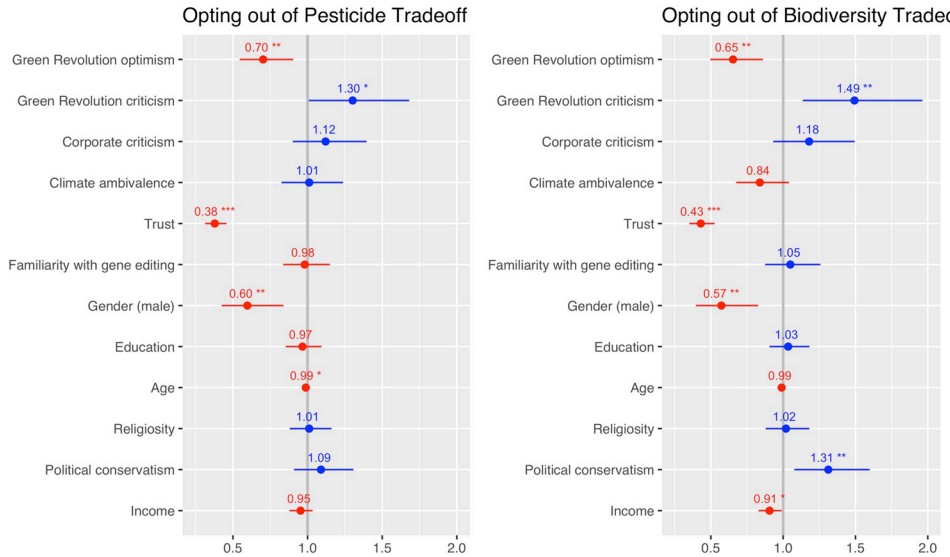

**Fig 3. Likelihood of 'opting out' of of gene editing trade-offs.** Plotted below are the results of a binomial logistic regression comparing participants who 'opted out' of the tradeoff, with those who preferred gene editing. Odds ratios represent the odds than an outcome will occur given a specific variable. Confidence intervals (2.5% to 97.5%) offer a range of plausible odds ratios for each of the independent variables. The significance codes for P values are: 0 '\*\*\*' 0.001 '\*\*' 0.01 '\*' 0.05. Participants were more likely to opt out of trade-offs with pesticides and biodiversity if they were critical of the Green Revolution. They were less likely to opt out if they were optimistic about the Green Revolution, expressed higher level of trust, or were male. Participants were less likely to opt out of the pesticide tradeoff, specifically, if they were older, and the biodiversity tradeoff if higher income. They were also more likely to opt out of the biodiversity tradeoff if politically conservative.

urgency of climate change were more comfortable with climate-relevant applications of gene editing (DV1, wheat case) and also more inclined to prefer gene editing in the context of trade-offs (DV2). We discuss these findings in greater detail below.

Our results suggest that the salience of critiques of industrialized agriculture may be important predictors of attitudes towards gene editing (P1), a finding not previously demonstrated via quantitative studies of perceptions of genetic engineering. We found evidence to suggest that criticism of corporations, optimism about the Green Revolution, and criticism of the Green Revolution predicted—to varying degrees—attitudes towards gene editing. We interpret this finding as indicative of direct and latent concerns about the very nature of food systems. Our survey-based finding is consistent, perhaps unsurprisingly, with the small group qualitative results that inspired this study; these have demonstrated that systemic critiques are closely related to how people think about these technologies. For example, a recent study emphasized that concern about gene-edited foods is rooted in the perception that these techniques are not necessary, that other ways of addressing the climate-food security nexus are preferable, and that goals such as 'greater production' were not deemed valid reasons for utilizing genetic engineering [7]. Taken altogether, our findings suggest that critiques of food systems are important to consider—and that it is also possible to quantitatively measure the strength and pervasiveness of such views and their relationship with overall support for gene editing via larger-sample survey work. As this study has only examined views amongst Canadian and US participants, in future work it would be important to explore the pervasiveness of such critiques and views amongst populations elsewhere.

Also of interest is the finding that perceived *optimism* about the Green Revolution was a more significant predictor across explanatory variables as compared to *criticisms* expressed about the Green Revolution (including criticism regarding impacts to the environment, traditional crops, and inequality). One possibility for this observation is that specific concerns about the Green Revolution do not necessarily translate into opposition to gene editing. Some participants might have been comfortable with these technologies *despite* such concerns. Approximately half of participants (n = 757) expressed some degree of agreement with both the Green Revolution criticism scale and the Green Revolution optimism scale. Crucially, expressing optimism about the Green Revolution does not seem to imply that a participant does not also harbor criticisms about the Green Revolution. Regardless, those who were optimistic about the Green Revolution were more likely to be comfortable with gene editing than those who were simply uncritical of it.

Our results indicate support for our second proposition (P2)—that outright or absolutist opposition to gene editing may be predicted by views of food systems. While some may argue that a refusal to engage in tradeoffs around gene editing might be motivated by a sense of irrational disgust, we found that such refusals might also be motivated by criticisms of modern food systems. Thus, our findings suggest reason to be skeptical in future research about claims that genetic engineering attitudes are absolutist, immoveable, and/or rooted in 'irrational' concerns [53]. This possibility aligns with the findings of others who have emphasized that the perception of harm, rather than disgust, can explain opposition to genetic engineering [55]. In the future, research should aim to disentangle the many possible motivations behind such absolutist rejection of gene editing, which may include both affective measures such as disgust, but also may include objections as to the way that specific applications have been designed and the farming practices so associated (e.g., industrial farming in the case of hornless cattle).

Surprisingly, corporate criticism did not predict absolutist rejection of gene editing. One reason for this might be that we found *most* participants to be quite critical of corporations: it is possible that attitudes across participants in this study have become quite anti-corporate as a whole, and corporate criticism has thus become too widespread to stand as a defining feature

of comfort or discomfort with gene editing. Evidence supporting this interpretation is that mean and median scores for this scale were higher than for other scales (e.g., trust, and green revolution scales), with participants overall 'agreeing' with statements that were critical of corporations. This finding aligns with recent qualitative work [1, 7], which found that publics may take corporate power as an inevitable 'given' and so this did not inform their opinions on gene editing for the better or worse.

Finally, regarding P3, views of ambivalence about climate change as a problem did predict support for gene editing—in the case where the application offered specific climate adaptive benefits. Our results do not appear to suggest, however, that general climate-related attitudes predict overall acceptance of gene-editing. This finding is consistent with research demonstrating that the particular benefits in question matter when determining attitudes towards genetic engineering [80]. Further study of gene editing that investigates not just the techniques used (e.g., comparisons of GM vs. gene editing) but rather, the kinds of benefits introduced, is key. We found that those who viewed climate change as more urgent were inclined to prefer gene editing. This may suggest that there is a causal relationship between the two, but further research would be needed to assess this, and to rule out the possibility that other mechanisms are at play. We suspect that climate-urgent concerns may override other subtler risks or concerns in the gene editing case, but this bears further investigation.

Returning to our original aims, these results offer some initial support for the assertion that public views on gene editing link not only with demographic factors, degrees of trust, and level of concern for environmental and health risks, but also with perspectives on broader societal challenges facing food systems today. Indeed, technologies such as gene editing might be thought of not as isolated purchases that individual consumers grapple with, but perhaps, as votes for or against different types of socio-economic and ecological futures. It remains to be seen, however, what kind of agricultural, economic, social, and ecological systems gene-edited products will end up being part of: a continuation of today's industrial paradigm, which many consumers/publics oppose? A necessary solution to the growing challenge of rapidly progressing climatic change? Both, or neither?

We also offer a reflection on future directions for research on public perceptions of biotechnology. A growing body of scholarship in the area of responsible research and innovation (RRI) has increasingly offered insights into how biotechnology innovation might better attend to ethical and social considerations. Very few these insights have, as of yet, made their way into risk perceptions work. What would it look like to better incorporate questions of responsibility into the study of public perceptions of biotechnology? Two core tenets of RRI work are the need for recognition and consideration of the values and assumptions that underpin certain technological applications, and an emphasis on participatory inclusion of different perspectives and actors [42, 81–83]. Certainly, efforts to better incorporate societal values, as we have attempted here, may be helpful. However, as much perceptions work relies on quantitative surveys, RRI's emphasis on participatory, deliberative methods might be hard to translate. One way to work 'with' such surveys might be, as we have done in this paper, to aim to more fully incorporate societal values into the explanatory variables that they consider. Another is to employ mixed methods approaches and so a hybrid of insights from deliberative/small group studies and survey work [84]. A final suggestion is that perceptions work might aim to better capture a range of technological alternatives, following Hartley et al. [42]'s assertion that inclusion of different options and problem framings is important for responsible governance. Capturing 'alternatives' might look like the 'tradeoffs' that we presented in this survey, or it could take more novel methodological forms, such as different scenarios and choice architectures. Incorporating some of these considerations into conventional risk perceptions research might

be a starting place for producing findings more attentive to questions of responsibly governing gene-editing technologies.

## 5. Policy implications

Our findings suggest several policy recommendations for different actors. First, for funders and developers of gene editing applications, our findings suggest that climate-relevant applications of gene editing might need be prioritized over other uses of gene editing, which publics may deem less justifiable. These proponents should also focus their resources on innovation that supports more sustainable systems, rather than funneling resources into applications that merely sidestep existing and unsustainable systems (e.g., those used for monocultural operations as opposed to agroecological farming). While it is not clear that there is much appetite for a combination of gene editing and more agroecologically oriented sectors such as the organic industry [85], there may be still be other opportunities to pursue agroecologically aligned applications. Given the complexity of shifting to more sustainable food systems, an important way to do this is will likely be to prioritize the involvement of more diverse actors, particularly groups traditionally marginalized in the development of agricultural technologies, such as smallholders and farmworkers, as others have already recommended [41, 86].

Second, regulatory bodies might explore ways to facilitate or expedite the oversight of applications designated as either essential for climate purposes, or, which facilitate rather than evade transitions to more sustainable agricultural practices. The Norwegian Biotechnology Advisory Board has proposed such an approach to regulation, which differentiates between applications based not only on risk but also potential social benefit. While its operationalization has been critiqued as continuing to prioritize technological concerns over social, cultural or ethical issues [87], this example may still perhaps serve as a model for other jurisdictions looking to adapt such an approach [88, 89].

Lastly, social science researchers should continue to explore public attitudes towards a range of different applications, contexts and purported benefits, as our study has reiterated the importance of these in engendering different responses to engineered products. While this study has utilized a quantitative survey method in hopes of informing broader perceptions research, there is ongoing need for qualitative methods that can better access the motivations behind people's views, rather than infer the relevance or calculate the predictive power of a certain view.

## 6. Conclusions

These findings suggest that when developers aim to design technologies that align with the values and priorities of different groups, it will be important to consider not just technical details of the organism's modification, or the types and extent of modifications made. These proponents and their funders should also consider which crucial benefits (such climate adaptivity) applications offer. Furthermore, some people may oppose technologies that they perceive as perpetuating highly industrialized or corporate forms of agriculture. Will proposed applications be free of intellectual property right constraints, or supportive of agroecological farming practices? Ignoring such considerations in the design of novel gene-editing applications may risk alienating key groups who might be more responsive to new technologies if these more fundamental concerns were addressed.

## Supporting information

**S1 File. Perceptions of gene editing in agriculture.**
(DOCX)

## Author Contributions

**Conceptualization:** Sara Nawaz, Terre Satterfield.

**Data curation:** Sara Nawaz.

**Formal analysis:** Sara Nawaz, Terre Satterfield.

**Funding acquisition:** Terre Satterfield.

**Investigation:** Sara Nawaz.

**Methodology:** Sara Nawaz, Terre Satterfield.

**Project administration:** Sara Nawaz, Terre Satterfield.

**Supervision:** Terre Satterfield.

**Validation:** Sara Nawaz.

**Visualization:** Sara Nawaz.

**Writing – original draft:** Sara Nawaz.

**Writing – review & editing:** Sara Nawaz, Terre Satterfield.

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
