## [Decision Letter · Decision Letter 0]

9 Dec 2021

PONE-D-21-30464Persuasive promises or absolute opposition? Perceiving gene editing in light of climate change and & 'system-critical' thinkingPLOS ONE

Dear Dr. Nawaz,

Thank you for submitting your manuscript to PLOS ONE. After careful consideration, we feel that it has merit but does not fully meet PLOS ONE’s publication criteria as it currently stands. Therefore, we invite you to submit a revised version of the manuscript that addresses the points raised during the review process.

Please address each comment provided by the reviewers. You will find that both reviewers have raised issues about better framing of the paper's contributions and situating the paper in the theoretical literature. The reviewers also asked for clarifications in the paper.   Please ensure that your decision is justified on PLOS ONE’s publication criteria and not, for example, on novelty or perceived impact.

We look forward to receiving your revised manuscript.

Kind regards,

Yueming Qiu

Academic Editor

PLOS ONE

2. Please include your tables as part of your main manuscript and remove the individual files. Please note that supplementary tables (should remain/ be uploaded) as separate ""supporting information"" files.

5. Please amend either the title on the online submission form (via Edit Submission) or the title in the manuscript so that they are identical.

Reviewers' comments:

Reviewer's Responses to Questions

**Comments to the Author**

1. Is the manuscript technically sound, and do the data support the conclusions?

Reviewer #1: Yes

Reviewer #2: Partly

2. Has the statistical analysis been performed appropriately and rigorously? 

Reviewer #1: Yes

Reviewer #2: I Don't Know

3. Have the authors made all data underlying the findings in their manuscript fully available?

Reviewer #1: Yes

Reviewer #2: Yes

4. Is the manuscript presented in an intelligible fashion and written in standard English?

Reviewer #1: Yes

Reviewer #2: Yes

5. Review Comments to the Author

Reviewer #1: Article:

Surveyed Canadians and Americans over the age of 18 to gather data on the attitudes towards gene editing in agriculture. The hypotheses are that those who believe that climate change is a large issue will be more open to consider biotechnology, those who have negative views about corporations and the green revolution will have more negative views about gene editing and often opt out of discussing trade-offs. These were rated using two dependent variables, the first being comfort about three cases (quality, animal well-being, climate tolerance) and the second being assessing the trade-offs of using gene editing (less pesticides/increased biodiversity) with the option to opt out of discussing. The explanatory variables include demographic and attitudinal variables.

I found this article to be very interesting and led to some very useful information for a variety of readers. Such as the climate-concerned people trusted GE in the wheat case, but had less of a predictive model in the tomato and cattle cases, which both have direct implications to improve agricultural systems albeit the link to climate may not be as obvious. Critical views to corporations had less of an impact than hypothesized is also a valuable finding.

Comments

• Much of this research is based on that the potential for gene editing to be viewed in a slightly more optimistic light – was there any information provided or questions in the survey to ensure the surveyor understands the difference between GM and GE?

Reviewer #2: Overall, this paper provides a timely contribution to the literature on the responsible development of GE technologies in agriculture. The findings are not really a surprise – there not much new here, but it brings empirical data to the newer debate about GE in solving global challenges such as climate change and this is valuable. It also tells us about the importance of developing technology that addresses needs rather than only delivering economic value. This contribution is not well highlighted at the moment. In general, the paper needs more accuracy and situating in the theoretical literature. My suggestions for improvement are below:

The title needs to be changed. It does not well reflect the content of the paper.

The term ‘system critical’ is used throughout but it’s not explained and inconsistently kept in ‘’. I suggest this term is removed from the paper and instead the authors find another way to describe what they mean by this phrase. Even by the end of the paper, I don’t feel I know what the authors are referring to with it.

The front end of the paper needs to be rewritten to frame the paper, provide more accuracy about the current empirical realities, and situate the paper in an academic theoretical literature. The authors need to make clear what their contribution is and to which debate. Is this a paper that contributes to public perceptions literature or the responsible innovation literature? As it is, the paper seems to rush into the empirical work without taking to time and space to let the reader know why it matters theoretically and for policy. Why does this paper matter? What's the point you're trying to make?

The first paragraph needs more precision and accuracy. Who are the proponents? This sentence needs evidencing as it’s a crucial motivation for the paper – ‘Proponents also argue that the precision of edits and ability to produce non-transgenics render edited organisms less risky and less controversial than genetically modified organisms (GMOs), and thus more appealing to consumers’. Where the authors talk about proponents, I would encourage them to either name them or talk about the types of actors they are throughout the paper.

This paragraph frames the paper in terms of a dichotomy between rejection and acceptance which I think is rather unhelpful. The findings are much less dramatic and help us to better understand the connection between a number of variables which include the motivation behind the technology (to solve grand challenges and deliver public value or to serve corporate interests and deliver economic value), and the fit with values and ideas about the world people want to live in (industrialised agricultural systems). The paper’s contribution is valuable without having to sensationalise in terms of rejection/acceptance. More work needs to be done in this front end of the paper to ‘frame’ it in current policy realities about the failure of GE to solve the challenges it often claims to solve (i.e. feeding the world.) This paper might help frame the paper - Vanloqueren, G., & Baret, P. V. (2009). How agricultural research systems shape a technological regime that develops genetic engineering but locks out agroecological innovations. Research policy, 38(6), 971-983. This one would also help: Voegtlin, C., & Scherer, A. G. (2017). Responsible innovation and the innovation of responsibility: Governing sustainable development in a globalized world. Journal of business ethics, 143(2), 227-243.

The claim at the start of paragraph 2 is not really accurate. There are lots of papers that do just this. One example the authors should draw on is the Royal Society (UK) public dialogue on GE which is highly relevant to this paper and has not been acknowledged - https://royalsociety.org/topics-policy/publications/2018/genetic-technologies-public-dialogue/.

1.1, second para – this is not accurate. GM technologies had the potential to deliver a much broader range of products than they did – again, Vanloqueren & Baret may help here. Para 3, less clear that what? I think this is one of the strengths of the paper: ‘minimal research has demonstrated the relationship between views on climate change and views on biotechnology’. This is a key feature and much more convincing than claiming the paper is about ‘rejecting or accepting’ GE.

I’d like to see all the Hs listed to start with and then discussed in order before the sub-headings. How are the Hs connected? Iwould encourage the authors to use titles for these section headers rather than questions.

1.1 title – this question doesn’t reflect the H. I’m not sure you’re looking at how people are swayed are you?

The section numbering needs to be checked. Why 1.2.1 for H3?

Section question for 1.2.1 is unclear – how does this relate to H3? You’re not really looking at this are you? The first para is particularly unclear. What is being traded off? It seems more about opting out of decisions or avoiding them rather than trading off. You state ‘We seek to explore the degree of such moral opposition to gene editing’ but this isn’t what you do. This section needs significant clarification.

H2 and H3 – remove the term system critical and rephrase the Hs. H2 is rather confusing and consists of two sentences. It should be clear enough so a clarifying send sentence is not needed.

Methods - A lot depends on what information is provided to participants in terms of the green revolution. What was in the ‘tutorial’? More information needed here. How do you know their opting out isn’t just not fully understanding what the GR is?

Table 1 – This needs further clarification. It’s not clear what you are telling us here and why it’s important. Trust is not discussed before this table.

Figures – I could not see these very well - the quality was quite poor. I’m sure this will be sorted out in the printing.

The results and discussion are good. I would encourage the authors to push further in the discussion, taking the time and space to draw out the significance of the results for the literature. What do these results tell us about the theory and empirical reality? Without the theoretical framing at the start it’s a bit difficult at the moment, but I want to know why your results matter for the literature on responsible innovation, or public perceptions literature, for example.

The policy implications section is really good and I would recommend the authors expand this section considerably – this is one of the most important sections where you reflect on what your research means. However, the Norwegian Biotech Advisory Board reference is taken rather critically given the criticisms that have started to emerge. See -Kjeldaas, S., Antonsen, T., Hartley, S., & Myhr, A. I. (2021). Public Consultation on Proposed Revisions to Norway’s Gene Technology Act: An Analysis of the Consultation Framing, Stakeholder Concerns and the Integration of Non-Safety Considerations. Sustainability, 13(14), 7643.

6. PLOS authors have the option to publish the peer review history of their article (what does this mean?). If published, this will include your full peer review and any attached files.

Reviewer #1: **Yes: **Karen Massel

Reviewer #2: No

---

## [Author Response · Author response to Decision Letter 0]

27 Jan 2022

Please see our 'response to reviewers' document.

---

## [Decision Letter · Decision Letter 1]

7 Mar 2022

Climate solution or corporate co-optation? US and Canadian publics' views on agricultural gene editing

PONE-D-21-30464R1

Dear Dr. Nawaz,

We’re pleased to inform you that your manuscript has been judged scientifically suitable for publication and will be formally accepted for publication once it meets all outstanding technical requirements.

Kind regards,

Yueming Qiu

Academic Editor

PLOS ONE

Additional Editor Comments (optional):

Reviewers' comments:

Reviewer's Responses to Questions

**Comments to the Author**

1. If the authors have adequately addressed your comments raised in a previous round of review and you feel that this manuscript is now acceptable for publication, you may indicate that here to bypass the “Comments to the Author” section, enter your conflict of interest statement in the “Confidential to Editor” section, and submit your "Accept" recommendation.

Reviewer #1: All comments have been addressed

Reviewer #2: All comments have been addressed

2. Is the manuscript technically sound, and do the data support the conclusions?

Reviewer #1: Yes

Reviewer #2: Yes

3. Has the statistical analysis been performed appropriately and rigorously? 

Reviewer #1: I Don't Know

Reviewer #2: I Don't Know

4. Have the authors made all data underlying the findings in their manuscript fully available?

Reviewer #1: Yes

Reviewer #2: (No Response)

5. Is the manuscript presented in an intelligible fashion and written in standard English?

Reviewer #1: Yes

Reviewer #2: Yes

6. Review Comments to the Author

Reviewer #1: Thank you for making the appropriate changes o the manuscript. I am happy with the resubmission for approval.

Reviewer #2: This manuscript is much improved. Thank you to the authors for such detailed revisions. The paper is now a pleasure to read and will make a valuable contribution.

7. PLOS authors have the option to publish the peer review history of their article (what does this mean?). If published, this will include your full peer review and any attached files.

Reviewer #1: No

Reviewer #2: No

---

## [Editor Report · Acceptance letter]

11 Mar 2022

PONE-D-21-30464R1 

Climate solution or corporate co-optation? US and Canadian publics’ views on agricultural gene editing 

Dear Dr. Nawaz:

I'm pleased to inform you that your manuscript has been deemed suitable for publication in PLOS ONE. Congratulations! Your manuscript is now with our production department. 

Kind regards, 

on behalf of

Dr. Yueming Qiu 

Academic Editor

PLOS ONE